# Digital Dermatopathology of Scabies: HE-Compatible VIS–NIR Hyperspectral Imaging as a Label-Free Proof-of-Concept Approach

**DOI:** 10.3390/bioengineering13010016

**Published:** 2025-12-25

**Authors:** Maximilian Lammer, Matthias Schmuth, Paul Bellmann, Verena Moosbrugger-Martinz, Bernhard Zelger, Birgit Moser, Christian Wolfgang Huck, Rohit Arora, Miranda Klosterhuber, Johannes Dominikus Pallua

**Affiliations:** 1Department of Dermatology, Venereology and Allergy, Medical University Innsbruck, 6020 Innsbruck, Austria; maximilian.lammer@i-med.ac.at (M.L.); matthias.schmuth@i-med.ac.at (M.S.); verena.martinz@i-med.ac.at (V.M.-M.); birgit.moser@tirol-kliniken.at (B.M.); 2Private Dermatological Practice, Specialized in Dermatopathology, Mariahilfpark 1/6, 6020 Innsbruck, Austria; bernhard.zelger@gmail.com; 3Institute of Analytical Chemistry and Radiochemistry, University of Innsbruck, 6020 Innsbruck, Austria; christian.w.huck@uibk.ac.at; 4Department of Orthopaedics and Traumatology, Medical University Innsbruck, 6020 Innsbruck, Austria; rohit.arora@i-med.ac.at (R.A.); miranda.klosterhuber@i-med.ac.at (M.K.)

**Keywords:** label-free detection, reflectance spectroscopy, chitin, FFPE tissue sections, spectral segmentation, principal component analysis, digital microscopy, computational pathology

## Abstract

**Background:** Scabies, caused by *Sarcoptes scabiei* var. *hominis*, remains difficult to confirm histologically when parasites are sparse or fragmented. Conventional microscopy is particular but limited by small sample size, tissue destruction, and observer dependence. **Objective:** To evaluate visible–near-infrared hyperspectral imaging (VIS–NIR HSI) as a label-free optical method for detecting *S. scabiei* in human skin sections and to assess its compatibility with routine HE staining. **Methods:** Formalin-fixed, paraffin-embedded (FFPE) skin tissue from six patients with histologically verified scabies was analysed using VIS–NIR HSI (500–1000 nm). Unstained sections mounted on CaF_2_ substrates and parallel HE-stained slides were imaged. Spectral datasets were processed by principal component analysis and segmentation to distinguish mite structures from epidermal and dermal compartments. **Results:** The chitin-rich mite exoskeleton exhibited a reproducible reflectance slope in the near-infrared range (R_850_/R_550_ > 1.5), clearly separating parasite from host tissue (R_850_/R_550_ < 1.0). PCA confirmed consistent cluster separation across all cases (ΔPC ≈ 3.7 ± 0.2). These contrasts remained detectable in HE-stained sections, validating applicability to conventional slides. **Conclusions:** VIS–NIR HSI enables reliable, label-free detection of *S. scabiei* mites in both unstained and HE-stained human skin tissue. By combining morphological and biochemical information in a single modality, HSI represents a promising adjunct to digital dermatopathology and may improve diagnostic sensitivity in challenging or atypical cases.

## 1. Introduction

Scabies is a highly contagious parasitic skin disease caused by the human-specific mite *Sarcoptes scabiei* var. *hominis*. The infestation remains common worldwide and affects all age groups, from children to frail elderly patients in long-term care facilities. Clinically, scabies typically manifests with intense nocturnal pruritus and erythematous papules, nodules, vesicles, or burrows in characteristic locations, including the interdigital spaces, wrists, axillae, trunk, and genital region [1]. Outbreaks in institutions such as nursing homes, hospitals, or refugee shelters illustrate that scabies is not only an individual burden but also a relevant public health concern.

Despite these characteristic features, everyday diagnosis is often challenging. Pruritic lesions in typical distributions may closely resemble eczema, atopic dermatitis, insect bite reactions, or urticaria, and the history of contact is not always obvious or remembered [2,3]. As a result, misdiagnoses and delayed treatment are frequent, particularly in oligosymptomatic or modified scabies and in crusted forms in immunocompromised patients.

Conventional confirmatory tests focus on the direct detection of the parasite or its products. Microscopic examination of skin scrapings or material from visible burrows can demonstrate mites, eggs, or scybala and is therefore highly specific; however, the sensitivity is limited by sampling, operator experience, and mite density, and the procedure is labour-intensive [4,5]. Histological examination of punch biopsies can also reveal mites or mite fragments, but it is invasive and has modest sensitivity because the parasite is small, sparsely distributed, and easily missed in routine sections [6].

Importantly, the present work does not aim to establish in visible–near-infrared (VIS–NIR) hyperspectral imaging (HSI) as a standalone clinical diagnostic test for scabies. Rather, we position HSI as an adjunct to routine histopathology and digital slide review: an HE-compatible spectral overlay that can highlight chitin-rich parasite material in formalin-fixed, paraffin-embedded (FFPE) sections, thereby supporting the pathologist when mites are sparse, fragmented, tangentially sectioned, or morphologically subtle. This adjunctive concept is particularly relevant in scenarios where routine histology shows a compatible but non-specific pattern (e.g., hyperkeratosis with a dermal hypersensitivity-type inflammatory infiltrate including eosinophils) yet fails to provide direct parasitological confirmation due to sampling error or low mite density [6]. In such cases, an objective spectral contrast mechanism could increase diagnostic confidence, support rapid targeted re-review, and reduce observer dependence without altering established staining workflows.

Several bedside methods, such as the burrow ink test and adhesive tape stripping, are simple and inexpensive but have variable performance and remain strongly operator-dependent [7]. Dermoscopy increases diagnostic yield by enabling non-invasive visualisation of the mite and its burrow, yet it requires training, can be negative in pauciparasitic disease, and is not universally available [8].

More advanced optical techniques—including video dermatoscopy, reflectance confocal microscopy, and optical coherence tomography—offer high-resolution structural information and can visualise mites in vivo, but are largely confined to specialised centres due to cost and technical complexity [4].

Molecular assays, particularly PCR-based methods, can detect *S. scabiei* DNA in skin samples even when conventional microscopy fails, but they require dedicated laboratory infrastructure, standardised protocols, and careful sampling, and are not yet fully established in routine dermatology [2]. Taken together, these limitations mean that there is still no widely accepted, non-invasive diagnostic gold standard for scabies, and the balance between sensitivity, specificity, invasiveness, and practicality remains unsatisfactory in many clinical scenarios.

Against this background, there is growing interest in optical techniques that provide objective, quantitative contrast based on tissue biochemical composition. Recently, we demonstrated, using Fourier-transform infrared (FTIR) microscopy, that chitin-rich parasite structures in scabies lesions exhibit distinct spectral signatures compared with the surrounding epidermis and dermis in FFPE sections, establishing analytical proof-of-concept for label-free parasite detection at the tissue level [9,10]. Building on these findings, hyperspectral imaging (HSI) in the visible–near-infrared (VIS–NIR) range offers an alternative, reflection-based approach that combines morphological and spectral information in a single dataset.

HSI acquires a complete reflectance spectrum for each pixel in the field of view, resulting in a three-dimensional “hypercube” (x, y, λ) that captures spatially resolved spectral signatures of different tissue components [11,12,13]. Because optical properties such as absorption and scattering vary with molecular composition and microstructure, HSI can reveal subtle differences between tissue compartments without exogenous labels [14,15]. In biomedical research, VIS–NIR HSI has been used to distinguish benign from malignant lesions, identify tumour margins, and assess tissue perfusion or wound healing [11,12,16,17,18,19]. In dermatology, early applications include lesion classification, pigment and vascular mapping, and wound monitoring [18,20,21,22,23].

In the context of scabies, the chitinous exoskeleton of *S. scabiei* represents an attractive target for spectral imaging, as chitin is absent from human skin and may therefore generate a characteristic optical contrast against the protein- and lipid-rich host tissue [9,24,25,26,27,28,29]. We hypothesised that VIS–NIR HSI of FFPE skin sections could detect reflectance-based spectral markers associated with the mite exoskeleton and related structures and that these markers might persist even after routine haematoxylin and eosin (HE) staining. The present study, therefore, investigates whether VIS–NIR HSI can identify *S. scabiei* mites in unstained and HE-stained human skin sections and evaluates its potential as an HE-compatible adjunct to digital dermatopathology. Accordingly, the primary contribution of this work is technical feasibility and analytical proof-of-concept at the tissue-section level. Clinical utility and real-world workflow relevance are not addressed by the current study design and require subsequent prospective validation and workflow-oriented studies.

## 2. Materials and Methods

### 2.1. Patient Cohort

Skin biopsies from six patients with histologically confirmed scabies were included in this retrospective study. The cohort consisted of three men and three women with a mean age of 61.2 years (range 21–88 years). Biopsy sites comprised the thigh, gluteal region, lumbar area, abdomen, and two additional truncal or extremity locations that were not further specified in the original pathology reports. All samples originated from routine clinical care at the Department of Dermatology, Venereology and Allergy, Medical University of Innsbruck.

Clinical diagnosis was established based on typical symptoms, lesion morphology, and distribution, and subsequently confirmed by histopathology. For the present analysis, FFPE tissue blocks and corresponding slides were retrieved from the departmental biobank. Ethics approval was obtained from the institutional review board (EK 1011/2025), and all procedures complied with local regulations and the Declaration of Helsinki. Only cases with clear clinical and histological evidence of scabies were considered for inclusion. The same six FFPE cases were previously described in detail in the FTIR microscopy study by Lammer et al. [10], which analysed the same tissue blocks for mid-infrared spectral characterization.

### 2.2. Sample Selection and Histopathological Assessment

FFPE blocks and archived HE-stained slides were re-evaluated by two experienced dermatopathologists (B.Z. and M.S.). For each patient, one representative block with well-preserved parasitic structures was selected. From these blocks, serial sections were cut: one section was mounted on a CaF_2_ substrate (1.0 mm thickness; KORTH KRISTALLE GmbH, Altenholz, Germany) and left unstained for label-free HSI, while adjacent sections were placed on glass slides (1.0 mm thickness; Klinipath VWR International B.V., Amsterdam, The Netherlands) and stained with routine HE (MORPHISTO GmbH, Offenbach, Germany) for diagnostic confirmation and morphological correlation.

Routine histology consistently demonstrated *S. scabiei* mites or mite remnants, including chitinous exoskeleton fragments and scybala (faecal pellets), in the stratum corneum. In most cases, the epidermis showed marked reactive changes with compact hyperkeratosis, parakeratosis, and occasional intracorneal vesicles. The underlying dermis frequently contained a dense perivascular and interstitial inflammatory infiltrate composed of lymphocytes, histiocytes, and numerous eosinophilic granulocytes, in keeping with a hypersensitivity pattern. Figure 1 illustrates typical histopathological features from one case study at increasing magnifications, including intact mites embedded in the hyperkeratotic stratum corneum and the associated inflammatory reaction. Histopathological features were consistent with those previously reported for this cohort [10].

### 2.3. HSI

To investigate the utility of HSI as a novel approach for detecting scabies, we employed visible and near-infrared (VIS–NIR) HSI. Data were collected using the TIVITA Tissue System (Diaspective Vision GmbH, Pepelow, Germany), which enables spectral imaging between 500 and 1000 nm, and was connected to an Olympus PRO-VIS AX 70 microscope (Olympus Corporation, Ina Plant, Ina, Japan) via a standard C-mount. Before image acquisition, all infrared filters were removed to enable full-spectral capture within the target range. Imaging was performed under standardised illumination with a halogen light source providing uniform transmission through the stained sections. To correct for background noise and illumination variability, dark and bright references were recorded before each session, ensuring the reliability of the reflectance spectra [14,16].

For proof-of-concept analysis, representative FFPE skin sections from patients with confirmed scabies were selected. Analyses focused on regions of interest (ROIs) containing mites and adjacent host tissue, given that the current HSI setup does not support whole-slide imaging. This limitation restricts the assessment of intralesional heterogeneity. Sections were imaged at 20× magnification, capturing both morphological and spectral detail. Each dataset (“spectral cube”) comprised two spatial dimensions (x, y) and one spectral dimension (λ), allowing simultaneous assessment of structural and spectral characteristics [11,12]. To prevent light contamination, imaging was conducted in a shielded setup, and acquisitions were performed within minutes to minimise temporal illumination shifts. The whole process, including calibration, required ~10 s per image, highlighting the potential of HSI for rapid application in routine pathology workflows [30].

### 2.4. HSI Data Processing and Analysis

The collected hyperspectral datasets were analysed using the TIVITA Suite Tissue software (version 1.6.0.1; Diaspective Vision GmbH, Pepelow, Germany), which enables the generation of both true-colour images and false-colour (pseudo-colour) visualisations derived from spectral characteristics. For each region of interest (ROI) defined by prior histopathological annotation, mean spectral signatures were extracted. These signatures served as the foundation for differentiating tissue compartments and identifying regions consistent with scabies positivity.

Two complementary analytical strategies were employed. First, Principal Component Analysis (PCA) was used to reduce the high dimensionality of the HSI data while maintaining the majority of its variance. PCA was deliberately selected as an unsupervised, exploratory technique because the present proof-of-concept dataset is small, enriched for histologically positive scabies cases, and does not include negative or disease-control sections. Under these conditions, supervised classifiers (e.g., SVM, random forests, or deep learning) would be at high risk of overfitting and would not provide meaningful estimates of slide-level diagnostic performance. In contrast, PCA enables objective visualisation of intrinsic spectral variance and parasite–host separability and supports the derivation of interpretable candidate features that can later be used to seed supervised models once sufficiently large, balanced cohorts with appropriate controls are available. This unsupervised approach facilitated spectral classification, enabling separation of mite structures, host tissue compartments, and background based on their spectral signatures [31,32]. By combining the first three principal components, false-colour composite images were generated, providing a clear visualisation of tissue heterogeneity and highlighting spectral contrasts between distinct compartments.

The processed spectral maps, along with the false-colour images, were then exported for further comparison with conventional histopathological evaluation using hematoxylin and eosin (HE) staining, allowing for a direct correlation between spectral features and microscopic findings. A conceptual workflow of the HSI-based diagnostic process is illustrated in Figure 2.

## 3. Results

### 3.1. Distinct Spectral Signatures in Unstained CaF_2_ Sections

HSI reflectance spectra acquired from unstained CaF_2_-mounted skin sections showed precise, reproducible optical differences between *S. scabiei* mites and surrounding human tissue compartments (Figure 3). Across all six analysed cases, spectra extracted from manually annotated regions of interest (ROIs) showed highly consistent patterns.

The background signal from the CaF_2_ substrate remained flat across the 500–1000 nm range, confirming optical stability and serving as a reliable reference baseline. In contrast, the parasite’s chitinous exoskeleton exhibited a characteristic spectral fingerprint: a shallow reflectance trough around 550–580 nm followed by a continuous increase toward the NIR region (>700 nm). This upward slope reflects the low absorption and high scattering of chitin in the NIR range, representing a robust spectral marker that distinguishes parasite material from host tissue. Host compartments displayed class-specific but markedly different optical behaviour.

Nuclear regions showed reduced overall reflectance with a local minimum between 540–600 nm, consistent with light absorption by chromophores and dense nuclear material.Extracellular matrix (ECM) exhibited intermediate reflectance with a smooth, featureless curve that gently increased into the NIR range, reflecting its mixed collagen–elastin composition.Keratinised epidermis displayed high reflectance in the visible spectrum (500–650 nm) that gradually decreased at longer wavelengths, in line with strong scattering from keratin filaments.

Quantitatively, the reflectance ratio R(850 nm)/R(550 nm) averaged 1.62 ± 0.10 for parasite regions versus 0.92 ± 0.04 for host tissue (*p* < 0.001, n = 6), confirming a distinct positive NIR slope unique to the mite. These spectral contrasts were consistent across cases, independent of patient or biopsy site, demonstrating the robustness of the optical signature.

From an analytical standpoint, these findings indicate that VIS–NIR HSI can differentiate between parasite and host tissue based solely on their intrinsic biochemical composition—without the need for stains or labels—providing a reproducible quantitative spectral feature within mite-containing, histologically annotated ROIs. Because ROIs were selected based on visible mite structures and no negative or disease controls were included, the present study does not evaluate slide-level detection performance or diagnostic specificity; the spectral feature should therefore be interpreted as a candidate discriminator that requires controlled validation against negative and look-alike conditions.

### 3.2. Influence of HE Staining on Spectral Profiles

VIS–NIR HSI of HE-stained FFPE skin sections showed that exogenous dyes altered baseline reflectance profiles but did not obscure the characteristic spectral contrast of *S. scabiei* (Figure 4). The glass substrate contributed no additional features, confirming the accuracy of calibration and stable reference conditions.

The parasite’s chitinous exoskeleton retained its distinctive reflectance slope above 700 nm, although overall intensity in the visible region was slightly reduced due to overlap with the absorption bands of hematoxylin and eosin. Despite these dye-related effects, the upward NIR trend remained the most consistent spectral marker for parasite identification across all stained slides. Host tissue compartments exhibited the expected dye-specific modifications:Nuclei showed intense reflectance depression between 500–550 nm from hematoxylin absorption, followed by gradual recovery toward the NIR.Extracellular matrix (ECM) regions showed eosin-related flattening of the curve near 550–600 nm, with only moderate signal increase beyond 700 nm.Keratinised epidermis retained high reflectance in the short-visible range (500–650 nm) with an additional eosin-related dip near 560 nm, then decreased steadily at longer wavelengths.

Quantitatively, the mean reflectance ratio R(850 nm)/R(550 nm) for parasite regions remained >1.5, whereas host tissue values stayed <1.0, confirming that the diagnostic NIR slope persisted despite staining. The reproducibility of this ratio across all cases (ΔR ≈ 0.70 ± 0.08; *p* < 0.001) supports its potential as a simple quantitative discriminator.

Pathologically, these findings demonstrate that VIS–NIR HSI is compatible with conventional HE slides. Although stains compress spectral variance and reduce contrast magnitude, the parasite’s optical fingerprint remains detectable. This compatibility enables the retrospective analysis of archived histologic material and facilitates the seamless integration of hyperspectral overlays into existing digital pathology workflows.

### 3.3. Segmentation and Spatial Mapping

HSI enabled both spectral differentiation and spatial visualisation of *S. scabiei* within tissue sections, generating image-based outputs that combined biochemical and morphological information (Figure 5). In unstained CaF_2_-mounted sections, the reconstructed RGB image resembled brightfield microscopy but lacked chromatic cues from histologic stains. When processed using PCA, false-colour composite maps revealed distinct pseudo-colour clusters corresponding to specific tissue types. The chitin-rich mite exoskeleton appeared in contrasting pseudo-colours (typically red or green), clearly separated from the extracellular matrix and keratinised epidermis. PCA-based segmentation further refined this separation by grouping pixels with similar spectral characteristics, delineating parasite boundaries that closely matched their morphological outlines on reference images. In HE-stained sections, conventional RGB reconstruction provided a familiar histologic colour palette, with nuclei stained blue-violet and keratin or cytoplasm appearing pink to orange. Despite partial spectral compression from the dyes, PCA-derived false-colour maps continued to display recognisable spectral contrast between mite and host tissue. Segmentation maps successfully isolated parasite regions, although interclass boundaries were slightly less sharp compared with unstained preparations. Across all samples, spectral segmentation accurately localised mites and their remnants, confirming that VIS–NIR HSI can spatially resolve parasitic structures within complex skin architecture. The close concordance between segmented HSI outputs and histopathological morphology underscores the method’s diagnostic utility. From a digital pathology perspective, these segmentation results illustrate the type of spectral overlay that could be generated once whole-slide acquisition and automated tile-wise analysis are implemented. However, because segmentation was performed on preselected, mite-containing ROIs, this work does not demonstrate automated screening or case-level detection; evaluating such performance will require whole-slide data and inclusion of negative and disease-control cohorts.

### 3.4. PCA

PCA was performed on hyperspectral datasets to visualise and quantify the spectral separability of *S. scabiei* from host tissue and background (Figure 6).

In unstained CaF_2_-mounted sections, three distinct clusters were observed in principal-component (PC) space. Background pixels formed a compact, narrowly distributed group, confirming calibration stability and the substrate’s spectral neutrality. Human tissue pixels occupied a broader region due to heterogeneity in scattering and absorption, whereas parasite-derived pixels clustered separately along the first two principal components. The first component (PC1) explained approximately 66% of the total variance, primarily reflecting spectral trends in the 700–900 nm range, dominated by chitin scattering, while PC2 accounted for an additional 17%. This apparent displacement of the parasite cluster confirmed that the characteristic NIR reflectance slope served as a dominant diagnostic feature. In HE-stained sections, overall clustering remained evident, although partial overlap occurred between host tissue and background because of the absorption of hematoxylin and eosin in the visible region.

Nevertheless, the parasite cluster remained consistently segregated, indicating that its intrinsic spectral fingerprint persisted despite staining. The inter-cluster distance (ΔPC ≈ 3.7 ± 0.2 a.u.) was only moderately reduced compared with unstained sections, demonstrating that staining affected intensity but not class discrimination. From a diagnostic standpoint, PCA provides a robust and computationally efficient means of dimensionality reduction, allowing for the visualisation of complex hyperspectral data and the objective confirmation of parasite–host separability. The reproducible clustering pattern across all analysed cases supports the feasibility of incorporating PCA-based dimensionality reduction as an exploratory step to visualise intrinsic spectral variance and parasite–host separability within mite-positive ROIs, and to inform subsequent supervised approaches once larger, balanced cohorts with appropriate negative and disease controls are available.

### 3.5. Spectral and PCA Across Multiple Cases

To evaluate reproducibility across patients and preparation types, quantitative spectral and PCA metrics were extracted from eight independent hyperspectral datasets (Table 1, Figure 7). In all cases, hyperspectral segmentation consistently identified mite regions as spectrally distinct from epidermal and dermal compartments, confirming that the diagnostic spectral fingerprint of *S. scabiei* was robust to inter-case variability and staining differences.

Across the dataset, the reflectance ratio R(850 nm)/R(550 nm)—representing the NIR slope magnitude—averaged 1.62 ± 0.10 for parasite regions and 0.92 ± 0.04 for host tissue (*p* < 0.001). The mean difference (ΔR ≈ 0.70 ± 0.08) remained stable across all samples, independent of slide preparation. Similarly, the Euclidean centroid distance (ΔPC) between parasite and extracellular-matrix clusters in PCA space averaged 3.74 ± 0.15 a.u., markedly exceeding the intra-class dispersion (<0.6 a.u.). The first principal component (PC1) accounted for approximately 66% ± 2% of the spectral variance and primarily captured the 700–900 nm range, which is associated with chitin scattering.

Representative HSI outputs of *S. scabiei* mites across multiple cases are shown in Figure 8. Each row represents a different biopsy sample of human skin with confirmed scabies. The columns illustrate:Brightfield RGB image of the unstained or HE-stained section;Corresponding PCA-derived false-colour image highlighting spectral heterogeneity;PCA-based segmentation map delineating parasite boundaries (green/red) from surrounding keratinised and dermal regions (blue/orange).

The consistent spectral separation across patients demonstrates the robustness of HIS for distinguishing the chitin-rich mite exoskeleton from host tissue, independent of sample origin or staining method. These quantitative and visual results together confirm that VIS–NIR HSI provides highly reproducible optical discrimination of *S. scabiei*. The stability of both spectral ratios and PCA metrics across multiple independent cases supports their potential use as candidate standardised parameters for quantitative characterisation and as inputs to supervised classification models in future studies that include negative and look-alike conditions. Together, these results demonstrate that VIS–NIR HSI provides a robust, label-free, and diagnostically compatible method for identifying mite-associated structures in routine histopathologic specimens, establishing a solid foundation for future development and controlled validation of automated digital approaches.

These quantitative and visual results confirm that VIS–NIR HSI provides highly reproducible optical discrimination of *S. scabiei*. The stability of spectral ratios and PCA metrics across multiple independent cases supports their potential use as standardised parameters for automated, quantitative classification in digital pathology workflows. Across all analysed cases, VIS–NIR HSI consistently distinguished *S. scabiei* mites from human epidermal and dermal tissue based on their unique optical properties. The chitin-rich exoskeleton produced a reproducible near-infrared reflectance slope and discrete PCA cluster that remained detectable even after HE staining. Quantitative metrics—including the reflectance ratio R(850 nm)/R(550 nm) and PCA centroid distance (ΔPC)—showed minimal inter-case variability, confirming high measurement reproducibility. Spatial segmentation accurately delineated parasite structures in both unstained and stained slides, aligning closely with histologic morphology. Together, these results demonstrate that VIS–NIR HSI provides a robust, label-free, and diagnostically compatible method for identifying *S. scabiei* in routine histopathologic specimens, establishing a solid foundation for automated digital detection and further clinical validation.

## 4. Discussion

In this proof-of-concept study, we show that VIS–NIR HSI can reliably detect *S. scabiei* mites in FFPE human skin sections and that this detection is feasible in both unstained CaF_2_-mounted sections and conventional HE-stained slides. The chitin-rich exoskeleton of the mite produced a characteristic reflectance pattern with a steeper near-infrared slope than the surrounding epidermal and dermal tissue, which was captured by the reflectance ratio R(850 nm)/R(550 nm) and by PCA of the spectral hypercubes. Across all six cases, mite-associated regions formed distinct PCA clusters and showed consistently higher R(850 nm)/R(550 nm) values than host compartments, indicating robust spectral separability at the tissue level. Consistent with this feasibility aim, we emphasise that HSI is intended as an adjunct to conventional histopathology: it is expected to add value primarily in “diagnostically uncertain” biopsies where direct mite detection is challenging, rather than in cases with abundant, readily identifiable parasites.

An important observation is that these contrasts persisted—even though somewhat attenuated—after routine HE staining. This suggests that VIS–NIR HSI is compatible with established histopathological workflows and can be applied retrospectively to archived slides. In practical terms, this means that HSI does not necessarily require dedicated label-free preparation and can be integrated as an add-on to digital pathology for selected cases in which scabies is suspected but parasitic structures are few, fragmented, or challenging to recognise morphologically.

Our findings complement and extend previous work using FTIR microscopy on the same cohort [10], where chitin-associated absorbance bands in the 1000–1200 cm^−1^ region were shown to differentiate mite exoskeletons from surrounding skin compartments in FFPE sections [9]. While FTIR microscopy probes molecular vibrations in the mid-infrared range and provides particular spectral fingerprints, VIS–NIR HSI detects changes in reflectance and scattering across the visible and near-infrared spectrum. The latter approach offers lower spectral resolution but can be implemented with comparatively accessible optical hardware and is readily combined with brightfield microscopy and standard histological stains. Together, these techniques highlight that chitin-related optical properties of scabies mites can be exploited across different spectral domains and imaging modalities.

Conventional diagnostic methods for scabies—including skin scraping, dermoscopy, and histopathology—are limited by sampling error, operator dependence, and the often sparse distribution of mites [4,6,7,8]. Even when biopsies are performed, intact mites are rarely present in routine sections, and mite fragments may be inconspicuous in a background of hyperkeratosis, crusting, and inflammation [6]. By providing a quantitative, spatially resolved measure of optical contrast between parasite and host tissue, VIS–NIR HSI offers a complementary dimension that is independent of purely morphological recognition. In principle, this could increase diagnostic confidence in challenging cases, assist in teaching and quality assurance, and support automated parasite detection in digital workflows.

At the same time, several limitations of this work must be acknowledged. First, the sample size was intentionally small (n = 6) and limited to histologically confirmed cases of scabies. This design reflects the exploratory nature of the study and was chosen to ensure the availability of well-characterized, parasite-positive material for establishing analytical feasibility. However, the lack of negative controls—such as inflammatory dermatoses, pruritic eczema, or other ectoparasitoses—precludes any assessment of diagnostic specificity or sensitivity. Consequently, the present findings should not be interpreted as measures of diagnostic accuracy but rather as a technical demonstration of spectral separability under controlled conditions. Accordingly, the method demonstrated here should be understood as detecting chitin-rich mite-associated material within tissue sections, which is not equivalent to establishing a definitive clinical diagnosis of scabies in the absence of appropriate negative and disease-control comparators. Clinical interpretation, therefore, requires clinicopathological correlation. Future work should consequently include larger, prospectively collected cohorts with appropriate negative and disease controls to determine actual diagnostic performance and clinical applicability.

Second, our imaging protocol focused on manually selected regions of interest containing visible mite structures rather than performing whole-slide HSI. This approach is sufficient for demonstrating local spectral contrast but does not capture intralesional heterogeneity or quantify automated detection performance across entire sections. To scale the presented concept toward whole-slide imaging (WSI) and screening scenarios, two practical routes can be envisaged: (i) automated tile-wise HSI acquisition using a motorized stage, followed by standard pre-processing (dark/bright correction), mosaicking/stitching of hyperspectral tiles, and slide-level assembly; and (ii) a hybrid workflow in which routine brightfield WSI is used for rapid screening and to define candidate regions that are subsequently re-imaged by HSI (“HSI-on-demand”), thereby reducing acquisition time and data volume. In both settings, downstream analytics can be implemented in a tile-based manner (e.g., fixed-size tiles across the slide), using the spectral ratio and PCA-derived separability demonstrated here as initial features and subsequently extending to supervised machine-learning models once sufficiently large positive and negative cohorts are available. Achieving routine-compatible throughput will require optimisation of acquisition speed, data handling, and robust co-registration between HSI tiles and the corresponding WSI coordinate system. Technical developments toward faster acquisition and higher-throughput scanning will be required to enable whole-slide, routine-compatible HSI. Because ROIs were selected based on histologically visible mites, the current analysis demonstrates spectral separability only in regions already known to contain mite-associated material and does not test whether HSI could find mites in a screening setting. In addition, without negative controls or disease comparators, specificity cannot be assessed; therefore, the reflectance ratio/PCA separation should be regarded as a candidate spectral feature of chitin-rich parasitic structures rather than a validated diagnostic biomarker.

Third, data analysis in this study relied on supervised segmentation and PCA-based visualisation. This choice reflects the study’s feasibility aim and the limited availability of labeled control material; supervised digital pathology classifiers would require substantially larger, balanced datasets (including inflammatory mimickers and other ectoparasitoses) for robust training and validation. While these methods are well-suited for exploratory work, fully automated machine learning approaches—including support vector machines and deep learning—may further improve classification performance and scalability for large datasets.

Finally, all measurements were performed on FFPE sections under controlled laboratory conditions. Translation to fresh tissue or in vivo imaging would require dedicated instrumentation, careful consideration of additional confounders such as blood flow, pigmentation, and surface curvature, and rigorous validation in prospective clinical trials. Nevertheless, VIS–NIR HSI has already been explored in other dermatological applications and intraoperative settings, suggesting that non-invasive or minimally invasive deployment is technically feasible [11,12,16,18,20,21].

Despite these constraints, the present data provide a coherent proof-of-concept that VIS–NIR HSI can detect scabies-associated spectral signatures in both unstained and routinely stained human skin tissue. Future studies should extend this work to larger, prospectively collected cohorts with appropriate negative and disease controls, include other arthropod infestations as comparators, and integrate automated spectral classification into digital pathology workflows. If such studies confirm high analytical robustness and clinically valuable diagnostic performance, VIS–NIR HSI could become a useful adjunct to conventional histopathology and dermoscopy for diagnosing scabies and other parasitic skin diseases.

## 5. Conclusions

This proof-of-concept study demonstrates that HSI enables the reliable detection of *S. scabiei* mites in human skin tissue. Both unstained CaF_2_ sections and conventional HE-stained slides yielded diagnostically proper spectral signatures, with the parasite’s chitin-rich exoskeleton producing a distinct reflectance profile that allowed robust separation from host tissue compartments. Quantitative analysis confirmed higher accuracy and sensitivity in label-free preparations, but HE-stained sections also retained sufficient spectral contrast for mite identification.

These findings highlight the potential of HSI as a complementary tool to conventional histopathology, combining morphological and biochemical contrast in a single imaging modality. Beyond retrospective pathology, the visible–NIR spectral markers identified here motivate future studies assessing whether similar contrast mechanisms can be leveraged in broader diagnostic or non-invasive settings. However, clinical diagnostic discrimination, specificity, and automated screening performance are not established by this ROI-based proof-of-concept and will require validation in larger cohorts, including negative and look-alike conditions, ideally in whole-slide workflows. Larger validation studies and the integration of automated machine learning classification will be the critical next steps toward translating this approach into routine diagnostic workflows. In a diagnostic pathology context, VIS–NIR HSI could serve as an adjunct to digital slide evaluation, supporting automated parasite detection, improving reproducibility, and reducing observer dependence. Ultimately, integrating spectral imaging with conventional microscopy represents a step toward more quantitative, multimodal pathology, bridging optical biochemistry with routine histological practice. In its current proof-of-concept form, VIS–NIR HSI should be interpreted as an adjunct technique for highlighting chitin-rich parasitic structures in histological sections rather than as a standalone diagnostic test for scabies. Nevertheless, this study does not establish clinical diagnostic performance or workflow benefit in routine practice. Future work must therefore evaluate patient-level outcomes and operational feasibility in larger, prospectively collected cohorts with appropriate controls and in workflows that reflect whole-slide and screening scenarios.

## Figures and Tables

**Figure 1 bioengineering-13-00016-f001:**
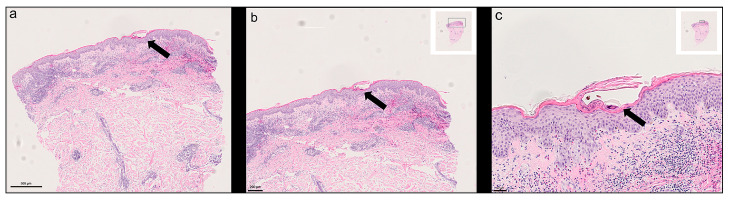
Histological features of scabies in HE-stained tissue sections. (**a**) Overview showing hyperkeratosis and epidermal hyperplasia with embedded mite structure; (**b**) Intermediate magnification revealing one mite in the stratum corneum and underlying inflammatory cell infiltrate; (**c**) High magnification of *S. scabiei* in the surrounding keratin layer. Scale bars are shown in the images—image taken from one of the study cases included in this report.

**Figure 2 bioengineering-13-00016-f002:**
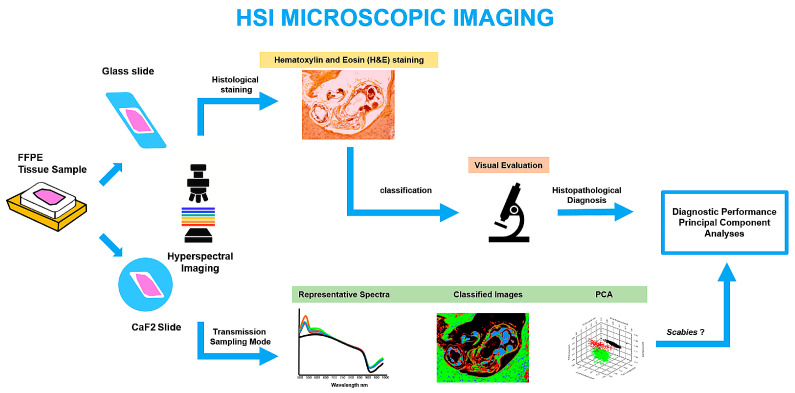
Workflow of HSI microscopic imaging. Both HE–stained and unstained FFPE tissue sections were analysed using HSI. Conventional histopathology relied on HE staining and visual evaluation, whereas HSI data enabled the extraction of representative spectra, generation of classified images, and application of PCA for dimensionality reduction and feature visualisation. This combined workflow allows complementary morphological and spectral tissue characterisation.

**Figure 3 bioengineering-13-00016-f003:**
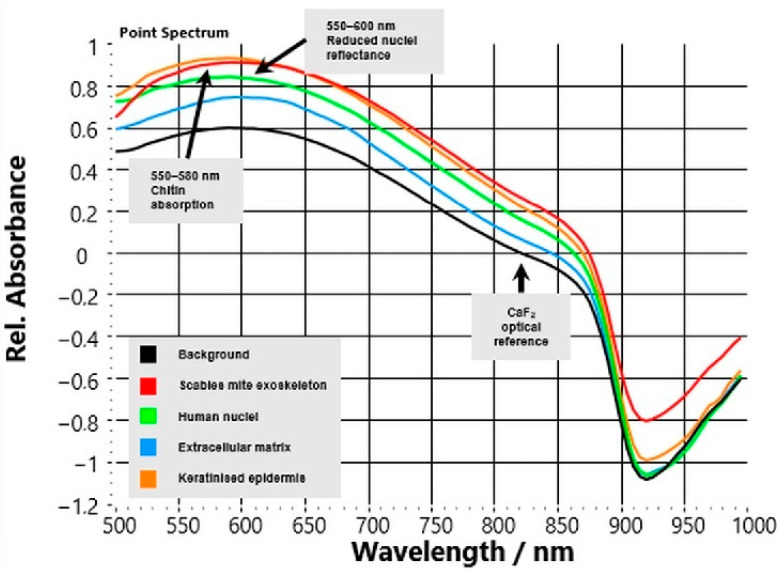
Mean hyperspectral reflectance spectra of unstained CaF_2_-mounted scabies sections. Spectra were extracted from ROIs representing background (black), mite exoskeleton (red), nuclei (green), extracellular matrix (blue), and keratinised epidermis (orange). On-plot annotations highlight the chitin-associated trough at 550–580 nm, reduced nuclei reflectance at 550–600 nm, and the CaF_2_ optical reference, illustrating spectral separability of mite material from host tissue.

**Figure 4 bioengineering-13-00016-f004:**
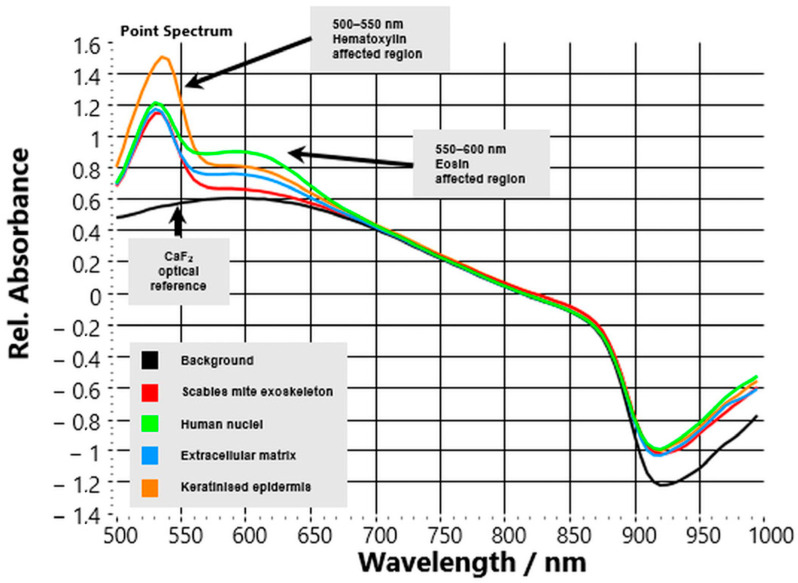
Mean hyperspectral reflectance spectra of HE-stained FFPE skin sections. Spectra were extracted from ROIs representing background (black), mite exoskeleton (red), nuclei (green), extracellular matrix (blue), and keratinised epidermis (orange). On-plot annotations indicate the dye-influenced wavelength ranges, including the hematoxylin-affected region (500–550 nm) and eosin-affected region (550–600 nm), while the CaF_2_ optical reference is shown for comparison. Despite dye-related spectral shifts, the mite-associated profile remains separable from host tissue, supporting feasibility of HSI on routine HE slides.

**Figure 5 bioengineering-13-00016-f005:**
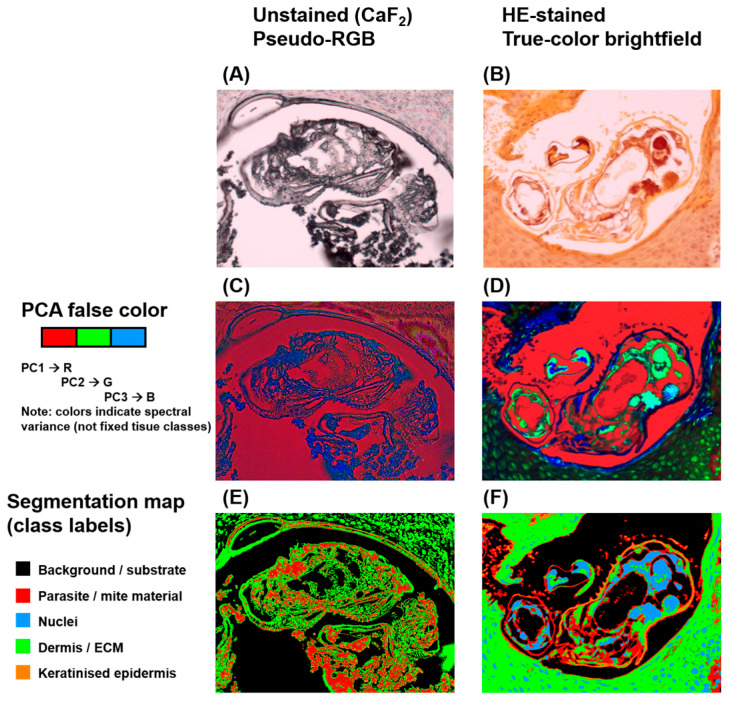
VIS–NIR HSI outputs of scabies mites in unstained CaF_2_-mounted sections (left column) and H&E-stained FFPE sections (right column). (**A**,**B**) Top row: pseudo-RGB band-composite reconstructions derived from selected spectral bands (not true-color imaging). Unstained CaF_2_-mounted sections appear predominantly gray because, in the absence of histological dyes, tissue contrast is dominated by scattering or absorption rather than chromophore-driven color. (**C**,**D**) Middle row: PCA false-color visualizations (e.g., PC1→red, PC2→green, PC3→blue), highlighting spectral variance and parasite–host contrast. (**E**,**F**) Bottom row: segmentation maps classifying pixels by spectral similarity; the color legend indicates the corresponding tissue/structure classes (parasite/mite-associated material, nuclei, keratinised epidermis, dermis/ECM, background/substrate) and is applied consistently across panels. In unstained CaF_2_-mounted sections, parasite–host separability is maximized due to preserved intrinsic spectral signatures; in H&E-stained slides, dye absorbance alters visible-range spectra, yet mite-associated structures remain distinguishable.

**Figure 6 bioengineering-13-00016-f006:**
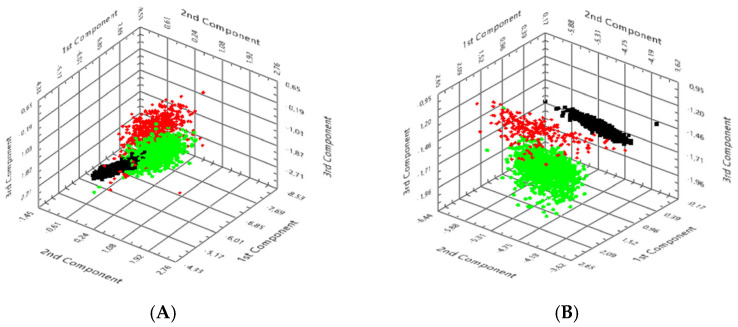
PCA of HSI datasets for unstained and HE-stained sections. (**A**) Unstained CaF_2_-mounted tissue sections: PCA scatter plots demonstrate three distinct clusters corresponding to background (black), human tissue (green), and scabies mite exoskeleton (red). Background pixels form a compact cluster, whereas mite pixels segregate clearly from human tissue along the first and second principal components, reflecting their unique chitin-derived spectral signatures. (**B**) HE-stained sections: while hematoxylin and eosin absorption reduces interclass variance and increases overlap between host tissue and background, the parasite cluster remains distinct. These results indicate that spectral fingerprints of scabies mites are robust to staining conditions, with clearer separability observed in unstained preparations.

**Figure 7 bioengineering-13-00016-f007:**
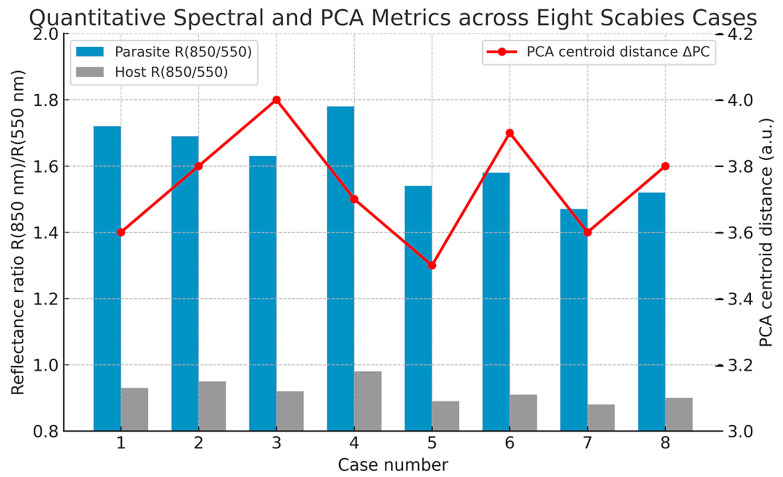
Quantitative spectral and PCA metrics across eight scabies cases. Bar plots show the mean reflectance ratio R(850 nm)/R(550 nm) for parasite (blue) and host tissue (grey), while the red line indicates the Euclidean centroid distance (ΔPC) between parasite and extracellular matrix clusters in PCA space. Across all datasets, the parasite shows markedly higher NIR reflectance ratios and stable PCA separability (ΔPC ≈ 3.7 ± 0.2 a.u.), confirming robust and reproducible discrimination of *S. scabiei* from human tissue based on VIS–NIR hyperspectral features.

**Figure 8 bioengineering-13-00016-f008:**
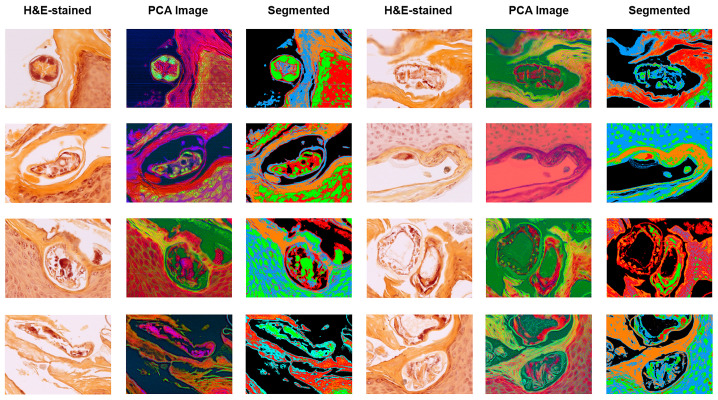
Representative HSI outputs of S. scabiei mites across multiple cases. Each row depicts different biopsy samples of human skin with confirmed scabies. Columns illustrate: (1) brightfield RGB image of the unstained or HE-stained section; (2) corresponding PCA-derived false-colour image highlighting spectral heterogeneity; and (3) PCA-based segmentation map delineating parasite boundaries (green/red) from surrounding keratinised and dermal regions (blue/orange). The consistent spectral separation across patients demonstrates the robustness of HSI for distinguishing the chitin-rich mite exoskeleton from host tissue.

**Table 1 bioengineering-13-00016-t001:** Quantitative spectral and PCA metrics across eight hyperspectral datasets.

Case	R(850 nm)/R(550 nm)—Parasite	R(850 nm)/R(550 nm)—Host (Mean)	ΔR-Ratio (Parasite–Host)	PCA Centroid Distance ΔPC (a.u.)	Variance Explained by PC1 (%)	Notes
1	1.72 ± 0.08	0.93 ± 0.05	0.79	3.6 ± 0.3	64.5	High SNR, distinct chitin slope
2	1.69 ± 0.06	0.95 ± 0.04	0.74	3.8 ± 0.4	66.1	Strong PCA separation
3	1.63 ± 0.07	0.92 ± 0.05	0.71	4.0 ± 0.5	65.9	Slightly lower reflectance amplitude
4	1.78 ± 0.09	0.98 ± 0.06	0.80	3.7 ± 0.3	67.4	Compact parasite cluster
5	1.54 ± 0.05	0.89 ± 0.04	0.65	3.5 ± 0.4	63.8	Dye absorption slightly reduces contrast
6	1.58 ± 0.07	0.91 ± 0.05	0.67	3.9 ± 0.4	64.2	Consistent NIR slope retained
7	1.47 ± 0.06	0.88 ± 0.04	0.59	3.6 ± 0.3	62.7	Mild overlap in PC2 direction
8	1.52 ± 0.05	0.90 ± 0.05	0.62	3.8 ± 0.3	63.3	Stable despite staining variability
Mean ± SD	1.62 ± 0.10	0.92 ± 0.04	0.70 ± 0.08	3.74 ± 0.15	65.9 ± 1.6	—

## Data Availability

The data presented in this study are available on request from the corresponding author. The data are not publicly available due to ethical restrictions.

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
