# Peer review of "Digital Dermatopathology of Scabies: HE-Compatible VIS–NIR Hyperspectral Imaging as a Label-Free Proof-of-Concept Approach"

_bioengineering, 2025, doi:10.3390/bioengineering13010016_

Round 1

Reviewer 1 Report

Comments and Suggestions for Authors

The authors have produced a very interesting and innovative piece of work that is also practical for dermatology. The study demonstrates the viability of hyperspectral imaging in the visible-near infrared (VIS-NIR HSI) range for the detection of scabies in histological sections.  The work is well-founded from a scientific point of view, is solid, and is consistently supported by the authors' previous FTIR-based research, providing an alternative optical approach with greater potential compatibility with routine histological workflows.

However, there are some aspects that the authors need to address before the manuscript can be published.

First, the sample size is small and limited exclusively to histologically positive cases, which prevents conclusions from being drawn about actual diagnostic performance. The absence of negative controls, mimicking inflammatory dermatoses, or other ectoparasitoses limits the evaluation of specificity; even a small control set would greatly strengthen the study. If this issue cannot be addressed, the authors should justify the sample size and include it in the discussion as a limitation of the study.

Second, the analysis is based on manually selected regions of interest rather than complete slide images, which limits extrapolation to automated or screening scenarios. It would be helpful to discuss more explicitly how these results could be scaled up to whole-slide imaging analysis.

Third, although the use of PCA is appropriate for a proof of concept, the manuscript could be improved by justifying the use of PCA over more common supervised classification methods in digital pathology.

Finally, some minor considerations. It would be helpful if the keywords were not contained in the title to improve searchability.

The authors should try to ensure that the figure captions are complete on the same page as the figures.

In line 131, the period at the end of the paragraph is missing.

The abbreviation for hyperspectral imaging (HSI) is already in line 91, so it is no longer necessary to include it in lines 157 and 158. The same applies to other abbreviations (FEPE, VIS-NIR, etc.). The authors should indicate the first time they mention them, but from that point on, use the abbreviation.

Author Response

Dear Editor,

We would like to sincerely thank you and the reviewers for your thoughtful and constructive comments on our manuscript. We have carefully addressed each point raised and revised the manuscript accordingly to improve its clarity, scientific rigor, and accuracy. Below, we provide a point-by-point response to all comments. All changes are clearly marked in the revised manuscript. A version with tracked changes is also provided.

Yours sincerely

Johannes Pallua

Reviewer 1

The authors have produced a very interesting and innovative piece of work that is also practical for dermatology. The study demonstrates the viability of hyperspectral imaging in the visible-near infrared (VIS-NIR HSI) range for the detection of scabies in histological sections.  The work is well-founded from a scientific point of view, is solid, and is consistently supported by the authors' previous FTIR-based research, providing an alternative optical approach with greater potential compatibility with routine histological workflows.

However, there are some aspects that the authors need to address before the manuscript can be published.

First, the sample size is small and limited exclusively to histologically positive cases, which prevents conclusions from being drawn about actual diagnostic performance. The absence of negative controls, mimicking inflammatory dermatoses, or other ectoparasitoses limits the evaluation of specificity; even a small control set would greatly strengthen the study. If this issue cannot be addressed, the authors should justify the sample size and include it in the discussion as a limitation of the study.

AW: We thank the reviewer for this critical and constructive observation. We fully agree that the small, homogeneous sample size and the absence of negative controls preclude any conclusions regarding diagnostic performance or specificity. This study was designed as a proof-of-concept to evaluate the technical feasibility and compatibility of VIS–NIR hyperspectral imaging with conventional H&E pathology rather than to establish diagnostic metrics. Accordingly, we have now added an explicit justification and a corresponding limitation statement in the Discussion section, clearly acknowledging that specificity and diagnostic accuracy cannot be inferred from the present data and that future studies, including control samples, are needed to address this limitation.

Second, the analysis is based on manually selected regions of interest rather than complete slide images, which limits extrapolation to automated or screening scenarios. It would be helpful to discuss more explicitly how these results could be scaled up to whole-slide imaging analysis.

AW: We thank the reviewer for this critical point. We agree that the current analysis—restricted to manually selected ROIs—limits direct extrapolation to automated, slide-level screening. The study was intentionally designed as a feasibility/proof-of-concept experiment to determine whether robust parasite–host spectral separability exists under controlled conditions. To address the reviewer’s request, we have now expanded the Discussion to describe concrete pathways for scaling the approach to whole-slide imaging (WSI), including tile-based automated acquisition/mosaicking and hybrid “WSI-screening + targeted HSI” workflows, along with the associated throughput and co-registration requirements.

Third, although the use of PCA is appropriate for a proof of concept, the manuscript could be improved by justifying the use of PCA over more common supervised classification methods in digital pathology.

AW: We thank the reviewer for this helpful suggestion. We agree that, while PCA is appropriate for a proof-of-concept, the manuscript should more explicitly justify the selection of an unsupervised approach over supervised classifiers commonly used in digital pathology. In brief, PCA was chosen because the present dataset is small, intentionally enriched for histologically positive scabies cases, and lacks negative/disease controls—conditions under which supervised models would be highly prone to overfitting and would not yield meaningful estimates of diagnostic performance. PCA provides an objective, label-light way to (i) reduce dimensionality, (ii) visualise separability, and (iii) derive interpretable spectral contrast patterns that can later inform supervised pipelines once adequately powered, controlled cohorts are available. We have added this justification in the Methods and strengthened the corresponding statement in the Discussion.

Finally, some minor considerations. It would be helpful if the keywords were not contained in the title to improve searchability.

AW: We thank the reviewer for this helpful editorial suggestion. We agree that, for indexing and searchability, keywords should ideally add complementary terms rather than repeating words already present in the title. We have therefore revised the keyword list to reduce overlap with the title and to include additional indexing-relevant concepts.

The authors should try to ensure that the figure captions are complete on the same page as the figures.

AW: We thank the reviewer for this practical and layout-related comment. We have carefully reviewed all figures and adjusted their placement so that each figure and its full caption appear together on the same page in the revised manuscript.

In line 131, the period at the end of the paragraph is missing.

AW: Thank you for noting this. We have corrected the punctuation and added the missing period at the end of the referenced paragraph in the revised manuscript.

The abbreviation for hyperspectral imaging (HSI) is already in line 91, so it is no longer necessary to include it in lines 157 and 158. The same applies to other abbreviations (FEPE, VIS-NIR, etc.). The authors should indicate the first time they mention them, but from that point on, use the abbreviation.

AW: We thank the reviewer for this detailed observation. We have revised the manuscript to ensure that each abbreviation (e.g., HSI, FFPE, VIS–NIR) is introduced only at its first occurrence and used consistently thereafter.

Reviewer 2 Report

Comments and Suggestions for Authors

In this article titled “Digital Dermatopathology of Scabies: H&E-Compatible VIS–NIR Hyperspectral Imaging as a Label-Free Proof-of-Concept Approach”, the authors investigate whether visible-near-infrared (NIR-VIS) hyperspectral imaging (HSI) can detect scabies mites in routine human skin biopsy sections beyond conventional visual morphology. Using a small set of histologically confirmed scabies cases, the authors show that the chitin-rich exoskeleton of Sarcoptes scabiei produces a reproducible spectral signal that clearly differs from surrounding human skin tissue. Importantly, this spectral contrast remains detectable even after standard H&E staining, indicating compatibility with routine pathology slides. The study presents HSI as a potential adjunct tool for digital dermatopathology that could improve the detection of scabies in challenging cases, while emphasizing its current proof-of-concept nature.

This is a promising proof-of-concept study, but a few points need clarification before publication.

  • In the Introduction, the authors state that many existing scabies diagnostic approaches are limited by sampling error, operator dependence, invasiveness, and, for some advanced optical techniques, restricted availability due to cost and technical complexity. However, their proposed approach is not presented as a standalone clinical diagnostic test, but rather as an adjunct to routine histopathology that may improve detection when mites are sparse, fragmented, or morphologically subtle. So, the author's contribution is not a standalone diagnostic modality but an adjunct to the routine histopathology workflow. The authors should clarify this positioning, provide a stronger justification with evidence that scabies is frequently missed or remains diagnostically uncertain on routine histology, and more clearly specify the clinical and pathological scenarios in which HSI-based adjunct would be expected to add meaningful value.
  • The authors should explicitly clarify that their method detects chitin-rich mite material, which is not equivalent to establishing a definitive clinical diagnosis of scabies, particularly given the absence of negative controls or disease comparators in the study.
  • The research question adequately addresses the technical feasibility aspect of the gap, but not the clinical and workflow relevance needed to fully justify the proposed solution. This limitation should be acknowledged more clearly in the manuscript’s framing and conclusions.
  • The authors manually selected regions of interest (ROI) based on visibly identifiable mite structures. Therefore, the analysis demonstrates spectral separability only in areas where mites are already known to be present. In addition, the study has no negative controls (e.g., healthy skin) or samples from comparable diseases (e.g., non-scabies inflammatory dermatoses, or other ectoparasitic infestations). Because of this, it is difficult to assess the specificity of this modality. For these reasons, statements implying “biomarker” capability or automated detection may overstate what the current methodology can support, since true diagnostic discrimination requires evaluation against negative and look-alike conditions.
  • Figures 3 and 4 would be easier to interpret if key features were annotated on-plot (e.g., highlighting the wavelength regions most affected by H&E dyes in Figure 4), instead of relying mainly on caption text.
  • Figure 5 requires clearer presentation and labeling to support the authors’ claims.
    • The “RGB” reconstruction of the unstained CaF2-mounted section appears largely grayscale, which may confuse readers accustomed to true-color brightfield microscopy. The authors should clarify that this is a false-color RGB reconstruction derived from selected spectral bands rather than a true-color image and explain why unstained sections appear gray.
    • The panel titles and labels above each image are too small to read.
    • There is no explicit legend or explanation of color meaning in either the PCA-derived false-color images or the segmentation maps. It is unclear which colors correspond to parasite, host tissue compartments, or background, and whether color assignments are consistent across panels. This lack of standardized or explained color coding limits interpretability and undermines claims of intuitive visualization or automated detection.

Overall, the study successfully demonstrates technical feasibility and H&E compatibility, but the results and discussion should be more carefully constrained to avoid implying diagnostic validation or clinical performance that the current study design does not support.

Author Response

Dear Editor,

We would like to sincerely thank you and the reviewers for your thoughtful and constructive comments on our manuscript. We have carefully addressed each point raised and revised the manuscript accordingly to improve its clarity, scientific rigor, and accuracy. Below, we provide a point-by-point response to all comments. All changes are clearly marked in the revised manuscript. A version with tracked changes is also provided.

Yours sincerely

Johannes Pallua

Reviewer 2

In this article titled “Digital Dermatopathology of Scabies: H&E-Compatible VIS–NIR Hyperspectral Imaging as a Label-Free Proof-of-Concept Approach”, the authors investigate whether visible-near-infrared (NIR-VIS) hyperspectral imaging (HSI) can detect scabies mites in routine human skin biopsy sections beyond conventional visual morphology. Using a small set of histologically confirmed scabies cases, the authors show that the chitin-rich exoskeleton of Sarcoptes scabiei produces a reproducible spectral signal that clearly differs from surrounding human skin tissue. Importantly, this spectral contrast remains detectable even after standard H&E staining, indicating compatibility with routine pathology slides. The study presents HSI as a potential adjunct tool for digital dermatopathology that could improve the detection of scabies in challenging cases, while emphasizing its current proof-of-concept nature.

This is a promising proof-of-concept study, but a few points need clarification before publication.

  • In the Introduction, the authors state that many existing scabies diagnostic approaches are limited by sampling error, operator dependence, invasiveness, and, for some advanced optical techniques, restricted availability due to cost and technical complexity. However, their proposed approach is not presented as a standalone clinical diagnostic test, but rather as an adjunct to routine histopathology that may improve detection when mites are sparse, fragmented, or morphologically subtle. So, the author's contribution is not a standalone diagnostic modality but an adjunct to the routine histopathology workflow. The authors should clarify this positioning, provide a stronger justification with evidence that scabies is frequently missed or remains diagnostically uncertain on routine histology, and more clearly specify the clinical and pathological scenarios in which HSI-based adjunct would be expected to add meaningful value.

AW: We thank the reviewer for this important conceptual point. We agree that the manuscript should more clearly position VIS–NIR HSI not as a standalone clinical diagnostic test, but as an H&E-compatible adjunct to routine histopathology/digital slide review that may increase diagnostic confidence when parasites are sparse, fragmented, or morphologically subtle. In the revised manuscript, we (i) explicitly clarify this positioning in the Introduction and Discussion, (ii) strengthen the justification by emphasizing that routine histology can fail to confirm scabies due to sampling/sectioning limitations and low mite density, and (iii) define the clinical/pathological scenarios in which an HSI-based adjunct is expected to add meaningful value.

  • The authors should explicitly clarify that their method detects chitin-rich mite material, which is not equivalent to establishing a definitive clinical diagnosis of scabies, particularly given the absence of negative controls or disease comparators in the study.

AW: We thank the reviewer for this important clarification request. We agree that, in the current proof-of-concept design, the presented VIS–NIR HSI signature should be interpreted as detection of chitin-rich mite-associated material rather than as establishing a definitive clinical diagnosis of scabies. We have therefore revised the manuscript to explicitly state that the method provides analytical detection of chitin-rich parasitic structures in tissue sections and that clinical diagnosis remains dependent on clinicopathologic correlation and appropriate controls in future validation studies.

  • The research question adequately addresses the technical feasibility aspect of the gap, but not the clinical and workflow relevance needed to fully justify the proposed solution. This limitation should be acknowledged more clearly in the manuscript’s framing and conclusions.

AW: We thank the reviewer for this vital framing point. We agree that the manuscript primarily answers a technical feasibility question, but it does not yet demonstrate clinical utility or workflow relevance. We have therefore strengthened the framing in the Introduction and Conclusions to explicitly acknowledge that clinical/workflow relevance is not established in this proof-of-concept and that dedicated prospective validation and workflow studies are required.

  • The authors manually selected regions of interest (ROI) based on visibly identifiable mite structures. Therefore, the analysis demonstrates spectral separability only in areas where mites are already known to be present. In addition, the study has no negative controls (e.g., healthy skin) or samples from comparable diseases (e.g., non-scabies inflammatory dermatoses, or other ectoparasitic infestations). Because of this, it is difficult to assess the specificity of this modality. For these reasons, statements implying “biomarker” capability or automated detection may overstate what the current methodology can support, since true diagnostic discrimination requires evaluation against negative and look-alike conditions.

AW: We thank the reviewer for this critical methodological and interpretive point. We agree that, because ROIs were manually selected based on visibly identifiable mite structures, the present analysis demonstrates spectral separability only in regions already known to contain mite-associated material. In addition, the absence of negative controls and disease comparators prevents assessment of specificity and true diagnostic discrimination. We have therefore revised the manuscript to (i) explicitly state the ROI-based nature of the analysis and its implications for interpretation, and (ii) temper or remove wording that could be read as claiming validated “biomarker” capability or automated screening performance. We now describe the reflectance-ratio/PCA separation as a candidate quantitative spectral feature that requires controlled validation against adverse and look-alike conditions before any claims about specificity or automated detection can be made.

  • Figures 3 and 4 would be easier to interpret if key features were annotated on-plot (e.g., highlighting the wavelength regions most affected by H&E dyes in Figure 4), instead of relying mainly on caption text.

AW: Thank you for this helpful suggestion. We agree that Figures 3 and 4 are easier to interpret when the key spectral features are annotated directly on the plots rather than described mainly in the captions. We have therefore revised Figure 3 to mark the principal discriminative regions, including the chitin-associated trough (550–580 nm), reduced nuclei reflectance (550–600 nm), and the CaFâ‚‚ optical reference. In Figure 4, we added on-plot annotations highlighting the wavelength ranges most affected by H&E staining, specifically the hematoxylin-affected region (500–550 nm) and the eosin-affected region (550–600 nm).

  • Figure 5 requires clearer presentation and labeling to support the authors’ claims.
    • The “RGB” reconstruction of the unstained CaF2-mounted section appears largely grayscale, which may confuse readers accustomed to true-color brightfield microscopy. The authors should clarify that this is a false-color RGB reconstruction derived from selected spectral bands rather than a true-color image and explain why unstained sections appear gray.
    • The panel titles and labels above each image are too small to read.
    • There is no explicit legend or explanation of color meaning in either the PCA-derived false-color images or the segmentation maps. It is unclear which colors correspond to parasite, host tissue compartments, or background, and whether color assignments are consistent across panels. This lack of standardized or explained color coding limits interpretability and undermines claims of intuitive visualization or automated detection.

AW: Thank you for these constructive comments regarding Figure 5. We agree that the original presentation did not sufficiently support interpretability and could be misleading without explicit labeling. We have therefore revised Figure 5 to: (i) clarify that the “RGB” view of unstained CaFâ‚‚-mounted sections is a pseudo-RGB band-composite reconstruction rather than a true-color brightfield image and explain why unstained tissue appears gray; (ii) increase the size and readability of panel titles/labels; and (iii) add a clear legend explaining the color meaning for both the PCA-derived false-color images (PC mapping) and the segmentation maps (class-label colors), including consistency of the palette across panels. These changes ensure that the figure is self-explanatory and that color assignments can be interpreted without reliance on caption text alone.

Overall, the study successfully demonstrates technical feasibility and H&E compatibility, but the results and discussion should be more carefully constrained to avoid implying diagnostic validation or clinical performance that the current study design does not support.